# Recapitulation of Retinal Damage in Zebrafish Larvae Infected with Zika Virus

**DOI:** 10.3390/cells11091457

**Published:** 2022-04-26

**Authors:** Adolfo Luis Almeida Maleski, Joao Gabriel Santos Rosa, Jefferson Thiago Gonçalves Bernardo, Renato Mancini Astray, Cristiani Isabel Banderó Walker, Monica Lopes-Ferreira, Carla Lima

**Affiliations:** 1Immunoregulation Unit of the Laboratory of Applied Toxinology (CeTICs/FAPESP), Butantan Institute, São Paulo 05503-900, Brazil; adolfo.maleski@esib.butantan.gov.br (A.L.A.M.); joao.rosa@esib.butantan.gov.br (J.G.S.R.); jefferson.bernardo@butantan.gov.br (J.T.G.B.); monica.lopesferreira@butantan.gov.br (M.L.-F.); 2Laboratory of Neuropharmacological Studies (LABEN), Post-Graduation Program of Pharmaceutical Science, Federal University of Sergipe, São Paulo 05503-009, Brazil; bandewalk@hotmail.com; 3Multipurpose Laboratory, Butantan Institute, São Paulo 05503-900, Brazil; renato.astray@terceiros.butantan.gov.br

**Keywords:** Zika virus, zebrafish, development, retinopathy, locomotor behavior

## Abstract

Zebrafish are increasingly being utilized as a model to investigate infectious diseases and to advance the understanding of pathogen–host interactions. Here, we take advantage of the zebrafish to recapitulate congenital ZIKV infection and, for the first time, demonstrate that it can be used to model infection and reinfection and monitor anti-viral and inflammatory immune responses, as well as brain growth and eye abnormalities during embryonic development. By injecting a Brazilian strain of ZIKV into the yolk sac of one-cell stage embryos, we confirmed that, after 72 h, ZIKV successfully infected larvae, and the physical condition of the virus-infected hosts included gross morphological changes in surviving embryos (84%), with a reduction in larval head size and retinal damage characterized by increased thickness of the lens and inner nuclear layer. Changes in locomotor activity and the inability to perceive visual stimuli are a result of changes in retinal morphology caused by ZIKV. Furthermore, we demonstrated the ability of ZIKV to replicate in zebrafish larvae and infect new healthy larvae, impairing their visual and neurological functions. These data reinforce the deleterious activity of ZIKV in the brain and visual structures and establish the zebrafish as a model to study the molecular mechanisms involved in the pathology of the virus.

## 1. Introduction

Zika virus (ZIKV) is an enveloped, positive-sense, single-stranded, mosquito-borne RNA virus with an icosahedral capsid belonging to the Flaviviridae family [1]. First discovered in Uganda, ZIKV has since disseminated widely throughout Africa, Asia, the Pacific Islands, and South and Central America, causing notable outbreaks in Micronesia (2007), French Polynesia (2013), and the Americas (2015–2016) [2].

Infected patients are typically asymptomatic; however, up to 20% of them may develop a mild, self-limiting symptom triad of generalized maculopapular rash, arthritis, and arthralgia following an incubation period of 3–14 days. Furthermore, ZIKV invades and persists in the central nervous system (CNS), provoking severe neurological diseases, affecting the ocular region with uveitis and conjunctivitis [3,4,5,6,7].

The first cases of ZIKV infection identified in the Northeast region of Brazil described as maculopapular rash manifestations associated with fever, conjunctivitis, and arthralgia [8] were followed by 5600 reported cases of congenital Zika syndrome due to vertical transmission characterized by severe microcephaly, thin cerebral cortices, retinal diseases, congenital contractures, and early hypertonia and extrapyramidal involvement. More recent studies have shown that 34–55% of children with microcephaly have ocular abnormalities, including chorioretinal atrophy with a hyperpigmented border and focal pigment at the macula, optic nerve abnormalities, and lens dislocation [9,10,11,12].

This posterior eye involvement dependent on the viral breakdown of the blood–retinal barrier or axonal transport along nerves [4,13,14] is followed by numerous other ocular manifestations leading to multiple deficits in visual function. Different in vitro and in vivo models have been used to model ZIKV infection of neural progenitors, which constitutes the pathogenetic key of brain and visual abnormalities [4], including bioengineered models such as human organoids [15].

The zebrafish, a teleost fish of the cyprinid family, fits the 3Rs philosophy perfectly [16], and experimentations are considered replacement or refinement methods [17]. In addition, the zebrafish in embryo stage is used as a confirmatory model of positive previously obtained results, thus having the ability to refine assessments [18,19]. The Brazilian legislations for the use of embryo/larval stages of zebrafish are in accordance with the European standards of animal welfare on animals used for scientific purposes [20], which dictate that only zebrafish up to 5 days post-fertilization (dpf) is considered a substitute for protected adults [21].

In addition to their morphological characteristics, such as the transparency of their embryos, ex uterus development, and the temporal separation of their immune system, they offer advantages such as a similar morphogenesis of the visual system composed of retinal epithelium, optic nerve, cornea, and iris, among other structures responsive to visual stimuli just 72 h after fertilization (hpf) [22,23]. Interestingly, the zebrafish has become a valuable tool for the study of human ophthalmological disorders. Many human ocular diseases, such as cataract, glaucoma, diabetic retinopathy, and age-related macular degeneration, have already been modeled in zebrafish [24,25].

Although only a few articles report naturally occurring viruses in adult zebrafish [26,27,28], the use of embryos to develop models of viral diseases has been demonstrated for other fish viruses [29,30,31,32] and also for mammalian viruses, such as herpes simplex virus type I [33,34], hepatitis B virus [35], hepatitis C virus [36,37], chikungunya virus [38], influenza A virus [39], Sindbis virus [40], vesicular stomatitis virus [41], norovirus [42], and dengue virus [43].

Zebrafish models of these human viral illnesses can supplement other animal models by providing the opportunity to directly visualize virus–host dynamics and to conduct genetic and chemical screens, which facilitate the development and testing of new antiviral therapies. Although our knowledge about ZIKV infection in the retina and its potential contribution to retinal pathology from murine models is still very limited [44,45,46,47,48,49,50], to date, no zebrafish retinopathy model has yet been recapitulated.

This work aimed to explore the potential of zebrafish as a model to recapitulate aspects of retinal lesions associated with ZIKV, focusing primarily on early morphological and behavioral phenotypes in developing zebrafish.

## 2. Materials and Methods

### 2.1. Zebrafish Husbandry

Wildtype (WT) AB strain adult zebrafish (<18 months old) from the International Zebrafish Resource Center (Eugene, OR, USA) were kept under the following standard conditions: temperature of 28 °C, pH 7, and light/dark cycle (14/10 h) in a flow-through system (ALESCO, Campinas, Brazil and Tecniplast, Varese, Italy) shelf using system water (60 μg/mL Instant Ocean sea salts). The fertilized embryos were visualized on an EZ4W stereomicroscope (Leica Microsystems, Cambridge, UK) and were classified [51] and incubated in E2 0.5× medium at 28 °C.

### 2.2. Zebrafish Anesthesia, Pigmentation Prevention, Dechorionation, and Euthanasia

Anesthesia was performed by submersion in E2 0.5× medium plus 0.4% tricaine (ethyl-3-aminobenzoate, #MS-222, Sigma Chemical Co., St. Louis, MO, USA) for 2 min at room temperature (RT). At the end of the experiments, euthanasia was carried out by immersion in 4% tricaine. Dead larvae were checked to ensure complete cessation of heartbeats, and then they were placed in a 10% bleach solution. To prevent pigmentation, embryos with 0 hpf were left in E2 0.5× medium with 1-phenyl-2-thiourea (PTU) at 0.00075% (#P7629, Sigma Chemical Co., St. Louis, MO, USA) for 6 h at 28 °C. When indicated, 24 hpf larvae were anesthetized and dechorionated by immersion for 5 min in pronase at 0.02 mg/mL (#P5147, Sigma Chemical Co., St. Louis, MO, USA).

### 2.3. Virus and Primary Infection

ZIKV^BR^ was donated by Dr. Pedro Vasconcelos, Instituto Evandro Chagas, Brazil [2]. Embryos at 0 hpf with 1 cell (one-cell stage embryo) mounted in the grooves of an agarose-coated plate (#16500100, Invitrogen, Carlsbad, CA, USA, EUA) were injected using an M205C stereomicroscope with ZIKV^BR^ samples (1 × 10^7^ PFU/mL) using a microneedle (#5242952008 femtotips 930000043 with 0.5–0.7 μm Eppendorf, Hamburg, Germany) coupled with an Injectman^®^ 4 pneumatic microinjector (Eppendorf, Hamburg, Germany) with approximately 2 or 3 nL in the yolk sac. After injections, embryos were incubated in E2 0.5× medium at 28 °C and analyzed for embryonic development after 3, 6, 12, 24, 48, and 72 h.

### 2.4. ZIKV Transmission

Next, 72 hpf zebrafish embryos infected at 0 hpf with ZIKV^BR^ (*n* = 12) were killed and lysated in 100 μL 1× Danieau solution (17.4 mM NaCl, 0.21 mM KCl, 0.12 mM MgSO_4_·7H_2_O, 0.18 mM Ca(NO_3_)_2_, 1.5 mM HEPES with pH 7.2). Then, they were cleared of tissue debris by centrifugation at 2000× *g*, and the supernatant was collected (ZIKV solution). Healthy and anesthetized 72 hpf larvae were infected either by immersion in E2 0.5× medium containing 0.04% of the ZIKV solution or by microinjection into the yolk sac (2–3 nL). Once again, the injections were conducted using a microneedle coupled with the pressurized microinjector (Eppendorf, Hamburg, Germany). After both methods of infection, larvae were incubated in E2 0.5× medium at 28 °C and analyzed after 24 h.

### 2.5. RNA Isolation and qRT-PCR

Entire larvae were harvested at specified time points and used for ZIKV titration by absolute real-time quantitative PCR (qRT-PCR). Zebrafish were collected (8–10 per group) and frozen in 500 μL Trizol reagent (Life Technologies, Carlsbad, CA, USA) at −80 °C. RNA was extracted by homogenizing the collected larvae through a 25G 1^1/2^ in needle on a 1 mL syringe (3–4 times until complete dissociation). Following the manufacturer’s protocol, RNA was set aside, and residual DNA was digested with a TURBO DNA-free kit (#AM1907, Invitrogen). cDNA synthesis was performed using one microgram of total RNA and the SuperScript VILO cDNA Synthesis Kit (#11754050, Invitrogen, Waltham, MA, USA). The RT-qPCR was run using the ViiA7 Real-Time PCR System, and each reaction included 5 μL of cDNA, a primer set, and TaqMan Universal Master Mix II (#4426710, Invitrogen) or SYBR green PCR Master Mix (#A25741, Applied Biosystems, Waltham, MA, USA). The primers (Integrated DNA Technologies, Coralville, IA, USA) were designed to amplify the genes of ZIKV, antiviral immunity type I IFN (IFNφ1; IFNφ3; IFNφ4), inflammatory response (IL-1; TNF; IL-6; IL-8; IL-34), and antimicrobial enzymes induced during inflammation (iNOSa; iNOSb; NOX1 gp67^phox^; NOX2 gp91^phox^). The relative expression of each gene was determined by comparing the endogenous controls (EF-1a; GAPDH) using the 2−ΔΔCt method, and values are expressed as fold induction relative to the expression level in the control group. For the analysis of gene expression, genes with fold change ≥1.5 were considered differentially expressed. ZIKV and zebrafish gene primers are demonstrated in Appendix A.

### 2.6. Phenotype-Based Screening

Mortality was counted and annotated daily. The surviving 72 hpf larvae were anesthetized and aligned in a glass dish in the lateral position and photographed under an M205C stereomicroscope. The images obtained were used to measure (*i*) body length from the top of the head to the tip of the tail, (*ii*) dorso-ventral head height, and (*iii*) eye along the nasotemporal axis using ImageJ v.1.8.0_172 [52].

### 2.7. Zebrafish Locomotor Behavior Assessment

Locomotor activity was investigated by analyzing the swimming behavior of 96 hpf zebrafish larvae upon dark–light transition according to the modified method of Scott, Marsden, and Slusarski [53]. Infected or non-infected larvae (*n* = 20) were transferred to 96-well plates, with one larva per well in 100 μL of 0.5× E2 medium, and analyzed in a Zebrabox System (ViewPoint Life sciences, Lyon, France). The larvae were analyzed for a total of 32.5 min, consisting of 30 min of acclimatization in the light (Lux: 12%) followed by 5 cycles of 1 s in the dark (Lux: 0%) and 29 s in the light (Lux: 12%). Locomotor activity was quantified and analyzed using ZebraLab™ version 3.52 (Pisa, Italy) by ViewPoint. The mean speed was set to values between 1.8 and 4.0 mm/s, while any movement slower than 1.8 was considered inactivity and above 4.0 as agitated behavior. The total distance results were obtained by summing the distances moved while at medium and agitated speeds, and the total average speed by dividing the distance by the analysis time. The startle response was generated by the distance that the larvae moved second by second (*n* = 20) during the adaptation period, followed by the light stimulus as described above.

### 2.8. Histology of the Eyes

Histological analysis was performed as described by Ferguson and Shive [54]. Whole zebrafish larvae were rinsed with sterilized PBS, fixed in 4% formaldehyde in PBS overnight, and then dehydrated by washing them sequentially with 30–70% methanol (MeOH) diluted in PBST (50 mL 10× PBS, 1 mL 10% Tween 20, up to 500 mL volume with dH_2_O) for 10 min each at RT with shaking. The next day, the larvae were rehydrated by washing them sequentially with 70–30% MeOH diluted in PBST for 10 min each at RT, with two sequential 5 min washes with PBST at RT. After removing the PBST, samples were transferred to a 30% sucrose solution diluted with deionized water and maintained overnight. For cryosectioning, fixed larvae were mounted in OCT compound (Neg-50, Richard Allan), sectioned at 18 μm in a coronal position on a cryotome (Cryostat Leica CM1860), and processed for hematoxylin/eosin (H&E). All slides were examined with light microscopy at 40× magnification (Axio Imager A1, Carl Zeiss; Oberkochen, Germany). For each group of 12 larvae, 8 stained larva sections from 3 larvae were analyzed per slide in ImageJ software (U.S. National Institutes of Health, Bethesda, MD, USA) by measuring the thickness of the ganglion cell layer (GCL), inner plexiform layer (IPL), inner nuclear layer (INL), outer plexiform layer (OPL), outer nuclear layer (ONL), retinal pigment epithelium (RPE), and lens.

### 2.9. Statistical Analysis

All values are expressed as mean ± SEM, using 20–100 larvae per group. Experiments were independently performed two times. Parametric data were evaluated using analysis of variance, followed by the Bonferroni test for multiple comparisons. Differences were considered statistically significant at *p* < 0.05 using GraphPad Prism (Graph Pad Software, v6.02, 2013, La Jolla, CA, USA).

## 3. Results

### 3.1. ZIKV Infection of One-Cell Stage Embryo Induces Mortality and Abnormal Development, and It Triggers Antiviral Innate Immune Response

The objective of this work was to model ZIVK infection in zebrafish embryos, particularly inducing phenotypical and morphological changes during larval development, which, in turn, leads to retinopathy. We tested the susceptibility of one-cell stage embryos to viral infection by injection into the yolk and subsequent maintenance through 72 hpf at 28 °C.

First, we demonstrated that ZIKV successfully infected zebrafish larvae. The expression of viral RNA was quantified to show the replication of the human viral pathogen in zebrafish. We confirmed the viral number and replication of ZIKV using PCR quantification of RNA extracted from the entire larvae. Efficient viral transcription was observed from 3 hpf, with an important transcription increment at 12 hpf (4-fold) reaching a peak of 7.4-fold at 72 hpf (Figure 1A). 

Our data of quantitative reverse transcription PCR (RT-qPCR) confirmed that ZIKV induces robust antiviral responses in microinjected embryos after 72 hpf, as demonstrated by the increased expression of IFN-φ1 and IFN-φ3 in ZIKV-infected larvae (30- and 28-fold, respectively) compared with embryos from the negative control group (Figure 1B).

Next, as can be observed in Figure 1C, the antiviral response induced by ZIKV was characterized by the expression of inflammatory cytokines such as IL-1β, IL-6, and IL-34 (nearly 20-fold) and predominantly by TNF-α at 400-fold compared with embryos from the negative control group. In contrast, the expression of IL-8 was less prominent. In the zebrafish larvae, ZIKV triggered high mRNA levels of iNOSa, iNOSb, and NOX2gp91^phox^ (nearly 20-fold) but low mRNA levels of NOX1 (Figure 1D).

One measure of defense against a viral pathogen is survival of the host. The mortality of infected larvae occurred at 24 h post-injection and was 12%, reaching 16% at 72 h compared with embryos in the negative control group, which had 100% survival over 72 h (Figure 1E).

Next, to verify the neurotropism of ZIKV, we focused on analyzing the development of structures such as the head and eyes of the ZIKV-infected zebrafish 72 hpf larvae. We noticed a reduction in the head size of the larvae estimated as 18% compared with embryos in the negative control group. No changes in the length of the body or in eye size were observed after ZIKV infection (Figure 1F).

### 3.2. ZIKV-Infected Zebrafish Larvae Show Motor Disorders and Signs of Visual Impairment

The reduced brain size may be the result of reduced neural stem cell (NSC) proliferation during the zebrafish stage of development [51]. ZIKV is involved in the impaired survival of NSCs in the brain at early developmental stages, thereby affecting the number of NSCs and, later, determining the size of the brain [3,4]. 

To further investigate whether the decrease in head size also implied changes in the neurological and visual systems, we used a Zebrabox system to analyze the behavior of ZIKV-infected larvae at 96 hpf. In Figure 2A, we observed a reduction of 63% in the total distance moved by the ZIKV-infected larvae during the total analysis period. ZIKV-infected larvae presented a decrease in mean velocity (Figure 2B). Furthermore, the swimming pattern of the ZIKV-infected group demonstrated inertia when compared to embryos in the negative control group (Figure 2C).

The locomotor activity of the zebrafish larvae depends on the integrity of brain function, nervous system development, and visual pathways [55]. Although there is no noticeable reduction in eye size (Figure 1F), behavioral changes in the swimming patterns (distance traveled and speed) of ZIKV-infected larvae may be the result of internal defects in eye development and structure. In Figure 2D, less activity was recorded during the light period, followed by immediate and robust hyperactivity in locomotor behavior during the darkness period for larvae in the negative control group. In contrast to the healthy uninfected larvae, ZIKV-infected larvae did not move after light stimulation or even in the dark.

### 3.3. ZIKV Alters the Thickness of Retinal Cell Layers

Recently, Wang et al. [56] identified the integrin αvβ5 as an internalization factor for ZIKV required for infection, which increased expression in radial glia, astrocyte, microglia, and endothelial cells but not neurons correlated with cell tropism ZIKV [57]. Comparable to the human eye, the mature zebrafish retina is composed of three nuclear layers separated by outer and inner plexiform layers (OPL and IPL, respectively). Photoreceptor (rod and cones) cell bodies reside in the outer nuclear layer (ONL); the amacrine, bipolar, horizontal, and Müller glial cell bodies occupy the inner nuclear layer (INL), and the ganglion cell bodies are contained in the ganglion cell layer (GCL). Synapsis between these nuclear layers and retinal neurons occurs at the plexiform layers [22].

We investigated the ability of ZIKV to induce viral tropism-dependent retinal damage. For this, we proceeded to conduct histological analysis of eye tissues sectioned at 18 μm in a coronal position and H&E staining after 72 hpf of infection of one-cell stage embryos (Figure 3A). As shown in Figure 3B, the retina of ZIKV-infected larvae showed an increase of 13% in total thickness accompanied by a 24% increase in the thickness of the lens compared to the negative control group. When the retinal layers were evaluated, we observed that the infection did not change the thicknesses of the RPE, OPL, ONL, and GCL. In contrast, ZIKV-infected larvae showed a 32% increase in IPL thickness (control 5.71 ± 0.2 vs. ZIKV 7.52 ± 0.1) and a 24% increase in INL thickness (control 19.54 ± 0.1 vs. ZIKV 24.3 ± 0.1).

### 3.4. Zebrafish Larvae Serve as Vectors for ZIKV Replication

Once we demonstrated that ZIKV was able to induce morphological changes in the eye during embryonic development, our next step was to verify its ability to induce changes in visual capacity in 72 hpf healthy larvae.

For this, we prepared a viral solution composed of the macerate of 72 hpf larvae previously infected with ZIKV^BR^ at one-cell stage embryos, and we proceeded to infect healthy 72 hpf larvae via injection in the yolk sac or exposure in the E2 0.5× medium. After 24 h, we evaluated the larval zebrafish locomotor behavior, a powerful indicator of perturbations in the nervous system.

After accommodating the larvae in a 96-well plate (one larva/well), they went through a 30 min acclimation period followed by light stimulation cycles of 1 s in the dark and 29 s in the light, repeated five times. As seen in Figure 4, the two types of infection routes were able to induce locomotor changes in the infected larvae when compared to the control group, and these were expressed by increased total distance moved (Figure 4A), higher velocity developed (Figure 4B), and an altered swimming pattern characterized by hyper sensibility (Figure 4C).

The tests employing different lighting conditions showed that 72 hpf larvae infected by injection or an immersion bath traveled greater distances with strongly increased speed after light stimulation and mainly in the dark when compared to control larvae that were more active in dark rather than in light periods (Figure 4D).

Together, these results reveal that, unlike the paralysis and hypoactivity of the larvae that were infected at the one-cell stage, healthy 72 hpf larvae that were infected with the viral solution showed hyperactivity to lighting conditions. These results show a significant difference in the neurological changes triggered by ZIKV during embryonic development or in the larval stage of 72 hpf.

## 4. Discussion

Zebrafish is increasingly being utilized as a model system to investigate infectious diseases and to advance our understanding of pathogen–host interactions. In the present work, we recapitulated for the first time ZIKV infection using zebrafish embryos and larvae, especially the ability of ZIKV to infect head and eye structures during embryonic development, leading to developmental changes characterized by a reduction in head size and increased thickness of the retinal cell layers with consequent impairment of neurological and visual functions. We found that ZIKV isolated from infected embryos caused retinal impairment when introduced into a healthy organism.

In addition to negative-sense RNA viruses, infections with some positive-sense RNA viruses, including dengue and Zika viruses, are also detected by retinoic acid-inducible gene I (RIG-I); this, in turn, activates interferon regulatory factor 3 (IRF3) and IRF7, which, together with the transcription factor nuclear factor-κB (NF-κB), induce the expression of type I interferons (IFNs) and other inflammatory genes [58]. The interferon (IFN) system is crucial in the fight against viruses, and it is capable of controlling most, if not all, viral infections in the absence of adaptive immunity [59].

In our model, ZIKV successfully infected zebrafish embryos, inducing an antiviral immune response and abnormal development of the larvae. Viral replication was maintained in 3 dpf larvae after injection at the one-cell stage and triggered a typical antiviral response, with the production of type I IFN, IFNφ1, and IFNφ3, which are characteristic of the larval stage [60]. As in mammals, antiviral immunity in zebrafish is orchestrated by virus-induced IFNs, structurally similar to mammalian type I IFNs [61,62]. The antiviral response with the production of type I IFNs was sufficient to prevent high mortality rates and even excessive damage to the eye, since the infected larvae did not show microcephaly.

We also found that ZIKV-infected larvae show inflammatory mediator production, particularly those known to recruit neutrophils. IL-1β is a chemoattractant for neutrophils in zebrafish, and TNF-α is one of the early immune genes expressed at an early stage of infection in fish and has a key role in regulating inflammation [63]. A preferential effect of IL-6 on Th2 pathways in fish has been demonstrated [64]. Zebrafish has two distinct CXCL8 homologs (IL-8) expressed in leukocytes: Cxcl8-l1 and Cxcl8-l2. Both Cxcl8 genes are up-regulated in response to an acute inflammatory stimulus, and both are crucial for normal neutrophil recruitment to the wound and normal resolution of inflammation [65]. Pro-inflammatory cytokines drive the production of IL-34 by neurons in the brain and keratinocytes in the epidermis, which supports microglia and Langerhans cell growth and survival through its binding to the MCSF receptor (MCSFR, also known as CSF1R or CD115) [66].

Interestingly, our data, which demonstrate the expression of the duplicated inducible nitric oxide synthase genes (nos2a and nos2b) and IFNφ1 and IFNφ3, point to the involvement of both with infection-induced granulopoiesis, as confirmed by Hall et al. [67]. Thus, we hypothesized that, in response to ZIKV infection, zebrafish larvae produce mediators that act on granulopoiesis and others responsible for the systemic recruitment and targeting of neutrophils to the site of infection.

Other observations that can be made regarding the physical condition of virus-infected hosts include morphological alterations in surviving embryos. We found that ZIKV affected the neurological and visual systems of developing zebrafish embryos. We also showed changes in the locomotion and swimming pattern of ZIKV-infected larvae. Our data show that one-cell embryos that were injected into the yolk showed inertia after 96 hpf in contrast to the increased locomotor activity observed after 24 h in the 72 hpf larvae injected with or bathed in ZIKV solution. These behavioral pattern alterations of the infected larvae are classic signs of neurological and visual defects in response to infection.

Our data on the hypoactivity of the malformed zebrafish in both light and dark periods corroborate the findings of Padilla et al. [68], who showed that the reduced movement of activity represents a secondary effect of the malformation that generates failure in the visual ability to recognize the surrounding environment, while larval infection caused an anxiety-like behavioral change [69,70,71].

Next, we demonstrated that ZIKV induces abnormal eye development characterized by damage of the retina and lens with enlargement of the inner nuclear layer (INL), which contains one glial cell type, the Müller cell [22]. It has been described that damage to the zebrafish retina induces Müller glia to act as stem cells, generating retinal progenitors for regeneration. Müller glia cells respond to retinal injury by a gliotic response that is characterized by hypertrophy and increased Gfap expression [72]. Furthermore, Müller cells together with the extracellular matrix, vasculature and other neurons represent an appropriate niche where NSCs self-renew or differentiate [73].

Dysfunction of the visual capacity of the ZIKV-infected larvae may be associated with alterations in the refractive index of the lens, as its normal development has a major influence on the proper visual function. The increase in lens thickness induced by ZIKV may be the result of increased constituents, such as concentric layers of fiber cells, elongated lens fiber cells at the lens equator, and, consequently, higher concentrations of lens proteins that are linearly related to the refractive index [74].

Finally, changes in the thickness of the inner nuclear layer may be the result of the infiltration of immune cells that crossed the blood–retinal barrier. Activated leukocytes affect the normal growth of developing retina by the production of inflammation-related cytokines [75,76]. Additional investigations of the lens protein gradient or inflammatory leukocytes in the retinal environment are needed to fully elucidate ZIKV-induced retinopathy of zebrafish larvae.

Taken together, our data show that zebrafish is an excellent model for recapitulation of retinopathy induced by ZIKV infection during early embryonic development. We found that ZIKV replication during embryonic development triggered a strong antiviral and inflammatory innate immune response. The infection caused low mortality, but the embryos showed abnormalities in retinal development with consequent visual and locomotor dysfunction.

## Figures and Tables

**Figure 1 cells-11-01457-f001:**
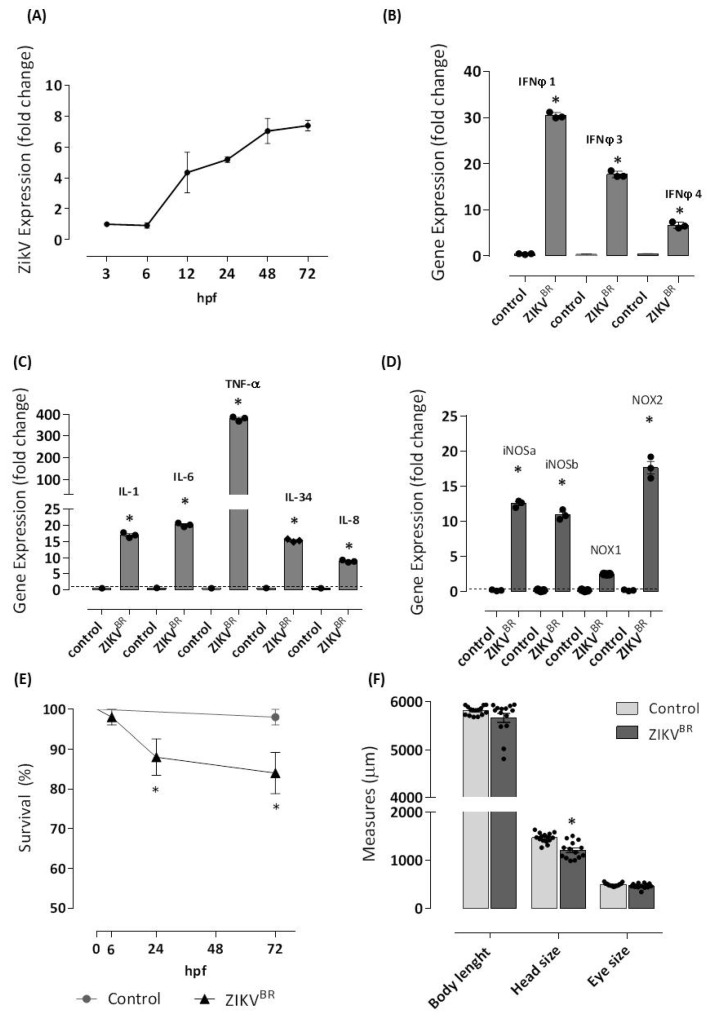
ZIKV infects one-cell stage embryo. The infection (*n* = 98) was performed by microinjecting embryos at 0 hpf with approximately 2 or 3 nL of ZIKV^BR^ (1 × 10^7^ PFU/mL, 2 a.p.) utilizing a M205C stereomicroscope coupled with an Injectman^®^ 4 microinjector. Another group (*n* = 83) was left without injection and formed the negative control group. Embryos were incubated in E2 0.5× medium at 28 °C, and viral RNA extracted from entire larvae was used for measurements via real-time quantitative PCR (qRT-PCR) after 3, 6, 12, 24, 48, and 72 hpf of the Zika viral load (**A**). Then, 72 hpf-infected larvae were sampled for qRT-PCR analysis of (**B**) IFNφ1, IFNφ3, and IFNφ4. cDNA was used in qRT-PCR reaction using primers specific for zebrafish, IL-1, TNF, IL-6, IL-8, and IL-34 (**C**), and iNOSa, iNOSb, NOX1, and NOX2 (**D**). The relative expression was normalized to the expression of EF-1a or GAPDH, and it is expressed as fold induction relative to the expression level in the control group (dotted line). For the analysis of gene expression, genes with fold change ≥1.5 were considered differentially expressed. ZIKV infection led to 14 and 16% of mortality (**E**), and a reduction of 17% of the head size was observed in ZIKV-injected larvae (**F**). Each bar represents the mean ± SEM. * *p* < 0.05 compared with the negative control group.

**Figure 2 cells-11-01457-f002:**
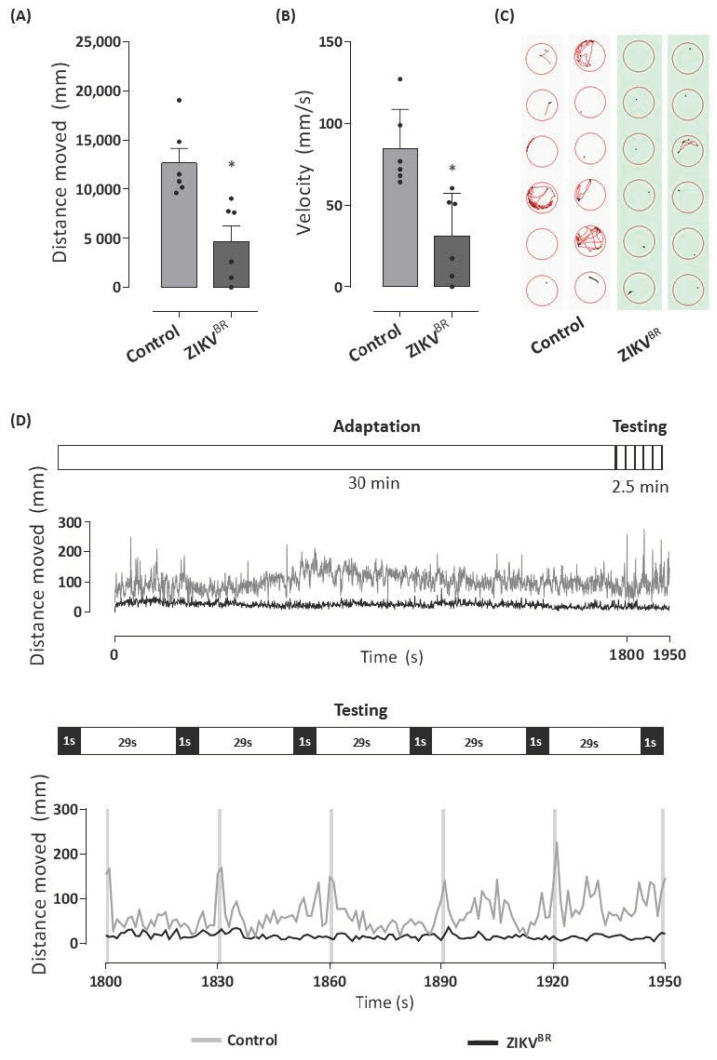
ZIKV infection of one-cell stage embryo changes locomotor activity behavior. The 96 hpf larvae infected with ZIKV^BR^ at 0 hpf or control group were distributed in a 96-well plate (1 larva/well), and the locomotor activity was analyzed in Zebrabox system. The larvae were exposed to an incubation period of 30 min in the light (lux: 12%) followed by the testing phase consisting of a dark stimulus (lux: 0%) for 1 s, alternated by 29 s in the light (lux: 12%) for 2.5 min to induce visual stimulation and the startle response. The total distance moved (mm) by each group during the testing period (**A**) and the average velocity of each group in mm/s are shown in (**B**). Graphical representation of larvae tracking during the testing phase (**C**). The lines represent the distance moved by the larvae at each second during the total experiment period with highlights of testing phase (**D**). Each bar represents the mean ± SEM. * *p* < 0.05 compared with the negative control group.

**Figure 3 cells-11-01457-f003:**
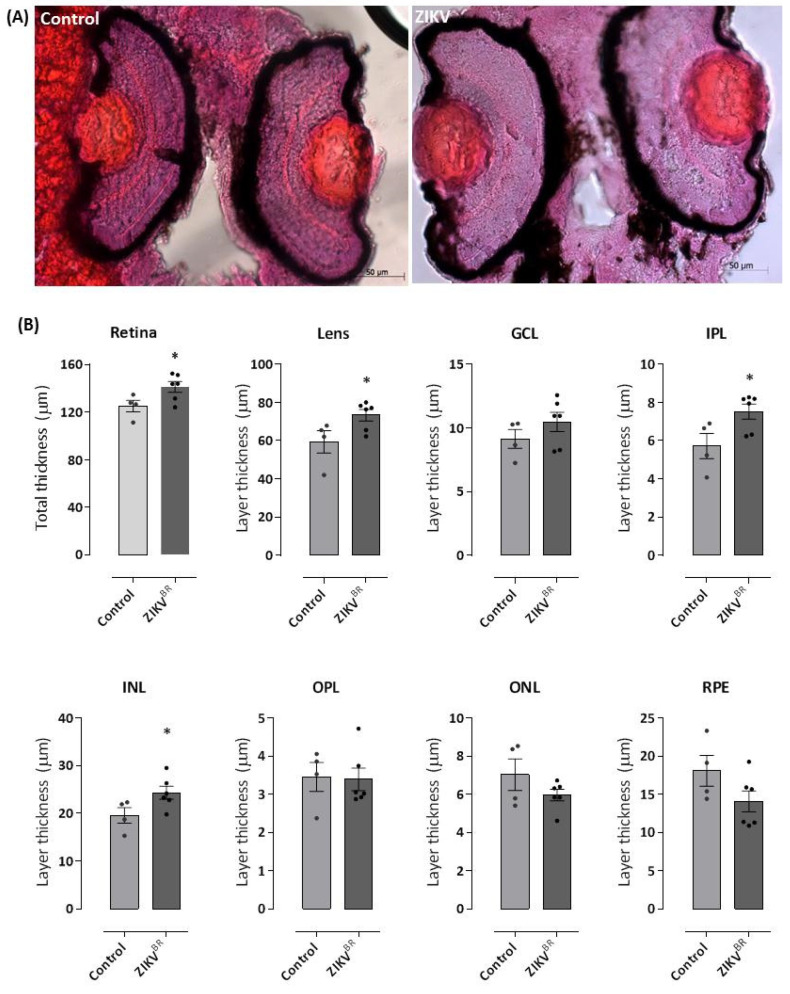
ZIKV causes an increase in the thickness of the zebrafish retina, especially in the lens and inner nuclear layer. The 96 hpf larvae infected with ZIKV^BR^ at 0 hpf or control group were fixed, dehydrated and rehydrated, incubated in 30% sucrose, and were mounted in OCT. Then, 18 μm-thick cryosections were stained with H&E. The images obtained under an optical microscope show an increased retinal layer thickness in ZIKV-infected larvae (**A**,**B**), which is most evident in the lens region. Red dots in the eyes of the control animals do not represent any morphological changes or developmental malformations. All measures were made in ImageJ software. GCL = ganglion cell layer; INL = inner nuclear layer; ONL = outer nuclear layer; IPL = inner plexiform layer; OPL = outer plexiform layer; RPE = retinal pigment epithelium. Each bar represents the mean ± SEM. * *p* < 0.05 compared with the negative control group.

**Figure 4 cells-11-01457-f004:**
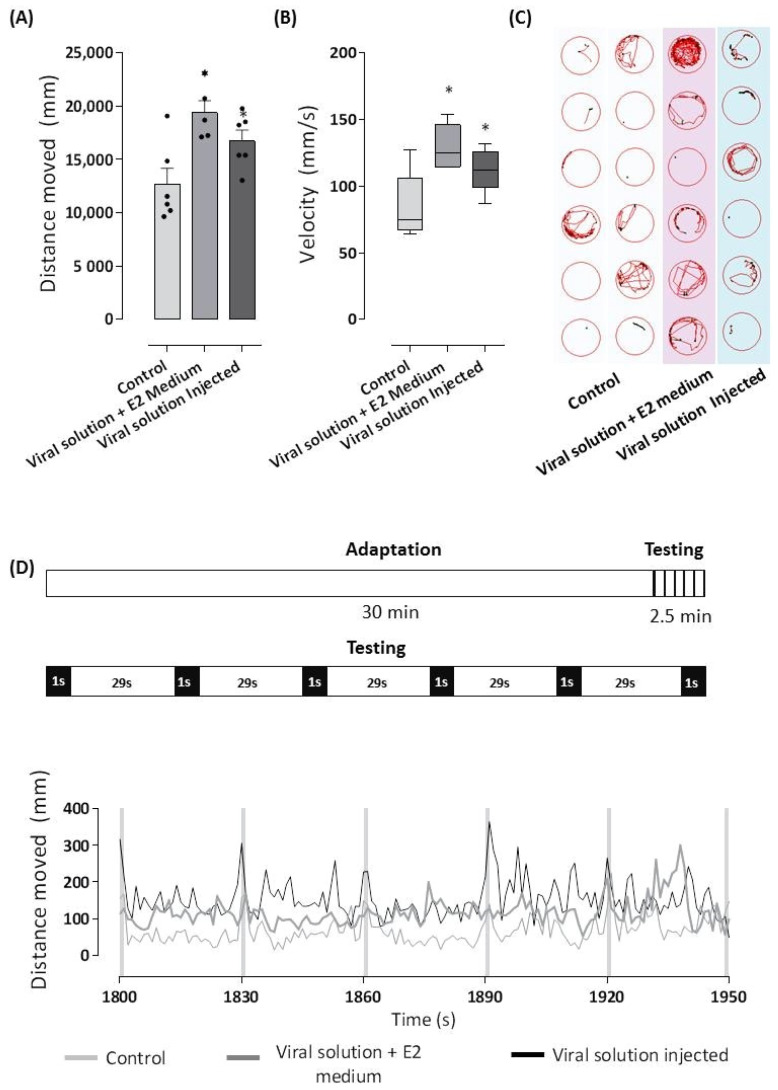
ZIKV replicated in zebrafish was able to infect healthy larvae. The supernatants obtained after centrifugation of the lysate from the 72 hpf ZIKV-infected larvae or control larvae were injected in 72 hpf healthy larvae into the yolk (2 nL) or diluted in E2 0.5× medium (0.04%). After 24 h, larvae were distributed in a 96-well plate (1 larvae/well), and the locomotor activity was analyzed in Zebrabox system. The larvae were exposed to an incubation period of 30 min in the light (lux: 12%) followed by the testing phase consisting of a dark stimulus (lux: 0%) for 1s, alternated by 29s in the light (lux: 12%) for 2.5 min to induce visual stimulation and the startle response. The total distance moved (mm) of each group during the testing period and the average velocity (mm/s) of each group are shown in (**A**,**B**). Graphical representation of the larvae tracking during the testing phase (**C**). The lines represent the distance moved by the larvae at each second during the total experiment period with highlights of testing phase (**D**). * *p* < 0.05 compared with the negative control group.

## Data Availability

Not applicable. Further inquiries can be directed to the corresponding author.

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
