# Peer review of "Recapitulation of Retinal Damage in Zebrafish Larvae Infected with Zika Virus"

_cells, 2022, doi:10.3390/cells11091457_

Round 1

Reviewer 1 Report

This study aims to use zebrafish embryos as a model for congenital ZIKV infection. In particular, the study focused on the effects of viral infection on vision and retinal development. While Zebrafish could be an extraordinary model to study host-pathogen interaction the main drawback of this study is the fact that actual infection and replication of the virus are not shown at any stage.

Replication could be shown by RT-PCR with negative and positive-strand primers or immunofluorescence with dsRNA antibodies. An infection could be proved by immunofluorescence with virus-specific antibodies. Also, plaque assays of the virus secreted to the media could be performed. Without these experiments, the observed effects could be non-specific.

Lines 102-103 please provide more details regarding the virus. Where was the GFP inserted? What was the source of the Brazil variant of the virus? ZIKVBR/IB what is IB?  

Figure 1: Can the authors supply any evidence that infection and replication of the virus did occur during this experiment? This might be a non-specific effect caused by activation of the immune system with non-replicating RNA. What was the efficiency of the infection?

Figure 3: size scale is missing. The figure is not clear. The same resolution should be shown for the control and ZIKV infected. Phase pictures should be shown for both also more than one embryo should be shown or a number of embryos in which there was ZIKV affinity for cells of the eye. Also, it will be helpful to show if the GFP signal increases from the infection time showing some indication that the virus is replacing.

Author Response

Reviewer 1

Comments and Suggestions for Authors

This study aims to use zebrafish embryos as a model for congenital ZIKV infection. In particular, the study focused on the effects of viral infection on vision and retinal development.

First of all, we appreciate you taking the time out to share your comments with us. We value and respect your opinion. We have already adjusted the manuscript, including additional changes as suggested.

While Zebrafish could be an extraordinary model to study host-pathogen interaction the main drawback of this study is the fact that actual infection and replication of the virus are not shown at any stage. Replication could be shown by RT-PCR with negative and positive-strand primers or immunofluorescence with dsRNA antibodies. An infection could be proved by immunofluorescence with virus-specific antibodies. Also, plaque assays of the virus secreted to the media could be performed. Without these experiments, the observed effects could be non-specific.

We include newly obtained PCR results that show virus kinetics in zebrafish embryos and cytokine profile at 72 hpf in ZIKV-injected embryos. In conjunction with IFN levels, this cytokine results show an embryonic response to viral replication.

Lines 102-103 please provide more details regarding the virus. Where was the GFP inserted? What was the source of the Brazil variant of the virus? ZIKVBR/IB what is IB?  

ZIKVBR BeH815744 (GenBank: KU365780) as was a courtesy of Dr. Pedro Vasconcelos was isolated from a human serum sample at Instituto Evandro Chagas. The data showing the tropism of zikv for the eye using the ZIKV-GFP were removed since when we asked our collaborator Renato Astray about the process of inserting the GFP into the ZIKV, we did not get an answer. We believe that histology data showing retinal abnormalities demonstrate the ability of ZIKV to alter embryonic development and visual function.

Figure 1: Can the authors supply any evidence that infection and replication of the virus did occur during this experiment? This might be a non-specific effect caused by activation of the immune system with non-replicating RNA. What was the efficiency of the infection?

We included data from PCR analyzes that prove the viral replication. In the revised manuscript, we present graphs of 1) viral kinetics, 2) anti-viral response, and 3) inflammatory response in embryos infected with 72 hpf

Figure 3: size scale is missing.

The figure is not clear.

The same resolution should be shown for the control and ZIKV infected. Phase pictures should be shown for both also more than one embryo should be shown or a number of embryos in which there was ZIKV affinity for cells of the eye.

  1. The quality of all images was adjusted with the inclusion of scales.

Also, it will be helpful to show if the GFP signal increases from the infection time showing some indication that the virus is replacing.

The data showing the tropism of zikv for the eye using the ZIKV-GFP were removed since when we asked our collaborator Renato Astray about the process of inserting the GFP into the ZIKV, we did not get an answer. We believe that histology data showing retinal abnormalities demonstrate the ability of ZIKV to alter embryonic development and visual function.

Reviewer 2 Report

The manuscript by Maleski et al describes the use of zebrafish larvae as a platform for studying Zika infection in the eyes. Overall, the manuscript is well-written and the conclusions drawn are supported by the data presented.

I have only minor comments.

I suggest authors improve the quality of the images by changing the color scheme used for graphing the data. The contrast of the greenish and gray color is hard to distinct in some figures. Lines are also too thinner for some figures. I also suggest showing all data points in the bar graphs.

Line 53: Readers might no be familiar with this concept. Please offer a brief explanation of the 3Rs proposed regulation before mentioning it in the text.

Lines 68-70: Please correct typos on the words ‘virus’

Line 203: Please correct typo on the word ZIKV

Author Response

Reviewer 2

Comments and Suggestions for Authors

The manuscript by Maleski et al describes the use of zebrafish larvae as a platform for studying Zika infection in the eyes. Overall, the manuscript is well-written and the conclusions drawn are supported by the data presented.

Thanks for reading the manuscript and providing interesting comments as feedback, we appreciate that. We considered the suggestions and have made significant improvements throughout the text.

I have only minor comments.

I suggest authors improve the quality of the images by changing the color scheme used for graphing the data.

  1. The quality of all images was adjusted with the inclusion of scales.

The contrast of the greenish and gray color is hard to distinct in some figures. Lines are also too thinner for some figures. I also suggest showing all data points in the bar graphs.

In all graphs, in addition to the group mean bar and SEM, individual plots were included. The colors and lines were suitable for better identification and visualization of the groups.

Line 53: Readers might no be familiar with this concept. Please offer a brief explanation of the 3Rs proposed regulation before mentioning it in the text.

  1. The concept of 3Rs has been explained in the text

Lines 68-70: Please correct typos on the words ‘virus’

  1. The words and sentences have been corrected

Line 203: Please correct typo on the word ZIKV

  1. The words and sentences have been corrected

Reviewer 3 Report

Broad comments

Novelty relies on, at least, for visual organ tropism in zebrafish model of ZIKV infection. The presentation of the results including the cartoons and graphs included in figures are really appreciable. The authors are aware of the limitation of not having already investigated the immune cells recruited after the productive infection of ZIKV, likely happening next to the eye. Despite I am not expert in zebrafish, from a cellular microbiology point of view, the article is qualified. I have only some comments written to improve the final outcome and potentiality of this manuscript after any publication.

Minor comments

  1. In Fig 3 FITc , the C should be capitalized
  2. In Fig 4 legend, IP = inner plexiform layer . it should be IPL like appearing in the image, please
  3. Even though zebrafish would be a good model to provide some embryological insights and transmission route during ZIKV in vivo infection, it should being advised to inform the reader that novel bioengineered models as human organoids can become useful to mimic the ZIKA pathogenesis (you may have a look at the following article as reference to be added https://doi.org/10.3390/pathogens10101233), because of viral elements binding specific human ligands/receptors.

  4. IFNphi1(a); IFNphi3(c); IFNphi4(d) please cheack (a)(c)(d), what are they? Where is (b)?
  5. In section 2.8 the abbreviation PBST is not extended. Is it PBS-Tween 20 as in Ferguson and Shive protocol? Please confirm and add final Tween concentration in the section
  6. Section 3.3 title should be italic
  7. Typo “FITc” on line 269. Also, in both legend and figure of figure 3on line 277
  8. In Fig 3 second column displays red spots. Are they overexposed points or does the red mean something or should be cited? Can be this clarified in the legend, please?
  9. At Line 341, please check “de” in the context of “The lines represents de the distance”
  10. Figure 5, “ZIKV replicated in zebrafish *were…”, I wonder if *”was”(singular) it is more appropriate if ZIKV is thought by authors as subject of the sentence.
  11. Figure 6 caption i.e. “Locomotor and visual analysis of zebrafish 96 hpf larvae infected with ZIKV. “should be formatted in bold
  12. On line 361, the authors wrote “we could suggest the co-participation of innate cells as neutrophils in the response to ZIKV”, but I would suggest to write something similar to "we hypothesize..." because Chikungunya virus share many things, but is different from ZIKV anyway.
  13. Viral genes and their regulation over-time were not analyzed collecting and lysing zebrafish eyes, despite PCR were available in the laboratory. Why? This is a limitation even fluorescent virus particles were shown at 72h (endpoint only).
  14.  

Author Response

Reviewer 3

Novelty relies on, at least, for visual organ tropism in zebrafish model of ZIKV infection. The presentation of the results including the cartoons and graphs included in figures are really appreciable. The authors are aware of the limitation of not having already investigated the immune cells recruited after the productive infection of ZIKV, likely happening next to the eye. Despite I am not expert in zebrafish, from a cellular microbiology point of view, the article is qualified. I have only some comments written to improve the final outcome and potentiality of this manuscript after any publication.

We appreciate you taking the time out to share your comments with us. We value and respect your opinion, and we have already adjusted the manuscript, including additional changes as suggested.

Minor comments

  1. In Fig 3 FITc , the C should be capitalized
  1. The words and sentences have been corrected

  1. In Fig 4 legend, IP = inner plexiform layer . it should be IPL like appearing in the image, please
  1. The words and sentences have been corrected

  1. Even though zebrafish would be a good model to provide some embryological insights and transmission route during ZIKV in vivo infection, it should being advised to inform the reader that novel bioengineered models as human organoids can become useful to mimic the ZIKA pathogenesis (you may have a look at the following article as reference to be added https://doi.org/10.3390/pathogens10101233), because of viral elements binding specific human ligands/receptors.
  1. We included other models in the text

  1. IFNphi1(a); IFNphi3(c); IFNphi4(d) please cheack (a)(c)(d), what are they? Where is (b)?
  1. The words have been corrected: antiviral immunity IFNφ1; IFNφ3; IFNφ4

  1. In section 2.8 the abbreviation PBST is not extended. Is it PBS-Tween 20 as in Ferguson and Shive protocol? Please confirm and add final Tween concentration in the section

PBST (50 mL 10x PBS, 1 mL 10% Tween 20, up to 500 mL volume with dH2O)

  1. Section 3.3 title should be italic
  1. The title has been corrected

  1. Typo “FITc” on line 269. Also, in both legend and figure of figure 3on line 277

The data showing the tropism of zikv for the eye using the ZIKV-GFP were removed since when we asked our collaborator Renato Astray about the process of inserting the GFP into the ZIKV, we did not get an answer. We believe that histology data showing retinal abnormalities demonstrate the ability of ZIKV to alter embryonic development and visual function.

  1. In Fig 3 second column displays red spots. Are they overexposed points or does the red mean something or should be cited? Can be this clarified in the legend, please?

Red dots in the eyes of the control animals do not represent any morphological changes or developmental malformations. The phrase was included in the legend figure.

  1. At Line 341, please check “de” in the context of “The lines represents de the distance”
  1. The words have been corrected
  1. Figure 5, “ZIKV replicated in zebrafish *were…”, I wonder if *”was”(singular) it is more appropriate if ZIKV is thought by authors as subject of the sentence.
  1. The words have been corrected

  1. Figure 6 caption i.e. “Locomotor and visual analysis of zebrafish 96 hpf larvae infected with ZIKV. “should be formatted in bold
  1. The caption was corrected

  1. On line 361, the authors wrote “we could suggest the co-participation of innate cells as neutrophils in the response to ZIKV”, but I would suggest to write something similar to "we hypothesize..." because Chikungunya virus share many things, but is different from ZIKV anyway.
  1. The sentence has been rewritten for better text flow

Viral genes and their regulation over-time were not analyzed collecting and lysing zebrafish eyes, despite PCR were available in the laboratory. Why? This is a limitation even fluorescent virus particles were shown at 72h (endpoint only).

The investigation of viral replication and cytokine levels was completed while the manuscript was on revision. We have included data from PCR analyzes that support viral replication. In the revised manuscript, we present graphs of 1) viral kinetics, 2) antiviral response, and 3) inflammatory response in 72 hpf infected-larvae

Round 2

Reviewer 1 Report

We include newly obtained PCR results that show virus kinetics in zebrafish embryos and cytokine profile at 72 hpf in ZIKV-injected embryos. In conjunction with IFN levels, these cytokine results show an embryonic response to viral replication.

IFN secretion is not a measure of replication. IFN is secreted also when synthetic dsRNA is transfected. Replication results are not convincing. I would probably not expect such an elevation in replication after such a short period of time (12 h) in such a system. A control with a mutation in the RdRp should be used. Which primers were used (not mentioned in the methods)?. Probably better to target the negative strand to prove replication.

Minor: 

The legend for figure 1a states that the time points are  3, 6, 24, 48 and 72 hpf the figure shows a 12 h time point?

Author Response

Cells

Manuscript ID: cells-1642998

Title: Recapitulation of retinal damage in zebrafish larvae infected by Zika virus

Authors: Adolfo Luis Almeida Maleski, Joao Gabriel Santos Rosa, Jefferson Thiago Gonçalves Bernardo, Renato Mancini Astray, Cristiani Isabel Banderó Walker, Monica Lopes-Ferreira, Carla Lima

Author's Reply to the Review Report (Reviewer 1) 2 round

We include newly obtained PCR results that show virus kinetics in zebrafish embryos and cytokine profile at 72 hpf in ZIKV-injected embryos. In conjunction with IFN levels, these cytokine results show an embryonic response to viral replication.

IFN secretion is not a measure of replication. IFN is secreted also when synthetic dsRNA is transfected. Replication results are not convincing. I would probably not expect such an elevation in replication after such a short period of time (12 h) in such a system. A control with a mutation in the RdRp should be used. Which primers were used (not mentioned in the methods)?. Probably better to target the negative strand to prove replication.

Infection kinetics, such as viral gene expression by PCR quantification of viral DNA or RNA, and viral burden when homogenates from virus-infected zebrafish were shown to infect healthy larvae represent an accurate measure in zebrafish and other animal models of viral infections and compared to human viral infections (Goody, Sullivan, & Kim, 2014; Sullivan et al., 2021). According to Robert Koch’s postulates to determine whether a microorganism is the cause of a disease, we found ZIKV in abundance in 72 hpf larvae suffering from the disease (Fig. 1A), but not be in healthy organisms, and isolated ZIKV from a diseased organism caused retinal damage when introduced into a healthy larvae. We observed the emergence in ZIKV-infected larvae of a protective immune response capable of limiting its spread (Fig. 1B, 1C, and 1D).

Goody MF, Sullivan C, Kim CH. Studying the immune response to human viral infections using zebrafish. Dev Comp Immunol. 2014 Sep;46(1):84-95. doi: 10.1016/j.dci.2014.03.025.

Sullivan C, Soos BL, Millard PJ, Kim CH, King BL. Modeling Virus-Induced Inflammation in Zebrafish: A Balance Between Infection Control and Excessive Inflammation. Front Immunol. 2021 May 7;12:636623. doi: 10.3389/fimmu.2021.636623.

Fig. 1A: Viral RNA extracted from entire larvae was used for measured via real-time quantitative PCR (qRT-PCR) after 3, 6, 12, 24, 48 and 72 hpf the Zika viral load (A).

Fig. 1B: 72 hpf infected-larvae were sampled for qRT-PCR analysis of IFNφ1, IFNφ3, and IFNφ4

Fig. 1C: 72 hpf infected-larvae were sampled for qRT-PCR analysis of IL-1; TNF; IL-6; IL-8, and IL-34

Fig. 1D: 72 hpf infected-larvae were sampled for qRT-PCR analysis of iNOSa; iNOSb; NOX1; and NOX2

We included all primers used in qRT-PCR in Supplementary Table 1.

Minor:

The legend for figure 1a states that the time points are  3, 6, 24, 48 and 72 hpf the figure shows a 12 h time point?

The time of 12 h was added to the legend.

Reviewer 2 Report

Lines 84-86: Please correct the error on the word virus. English: virus, not "vírus" (this is how you write it in portuguese).

Reviewer 3 Report

My comments were all addressed and I confirm my positive opinion on this article